# Autonomous Underwater Vehicle Navigation via Sensors Maximum-Ratio Combining in Absence of Bearing Angle Data

Vadim Kramar *, Aleksey Kabanov and Kirill Dementiev

Department of Informatics and Control in Technical Systems, Sevastopol State University,
299053 Sevastopol, Russia; kabanovaleksey@gmail.com (A.K.) mash.saigon.89@gmail.com (K.D.)
* Correspondence: kramarv@mail.ru

**Abstract:** This paper proposes a maximum-ratio combining sensor fusion scheme for using an extended Kalman filter in the underwater vehicle positioning task by means of communication devices (buoys) providing location information using a slant-range mechanism, inertial sensors, a Doppler velocity log, and a pressure sensor in the absence of bearing angle data. The parameter estimation methods for all navigation system components are described. The results of simulation modeling with corresponding quality metrics are presented. The outcomes were supported by conducted field experiments. The results obtained allowed us to obtain a position determination model for the underwater vehicle, which is still a relevant and complex task for seabed explorers.

**Keywords:** underwater navigation; inertial navigation; state estimation; sensors fusion; nonlinear filter

## 1. Introduction

Autonomous underwater vehicles (AUVs) are robots that are capable of moving underwater without operator assistance along trajectories determined according to the missions. AUV missions may include, for example, survey and exploration activities, including wreck search, geological exploration, seabed mapping, surveys of lengthy objects (submarine cables, pipelines), and environmental monitoring of water areas. At the same time, the AUV must have a long range [1] and, hence, solve complex navigation and positioning problems in both single-mission and group operations. The development of robust navigation strategies is necessary to ensure the proper execution of the AUV mission [2].

With the development of airborne and land-based navigation technologies, underwater navigation technologies have also evolved significantly [3–5]. However, due to the specificity of the underwater environment, there is still a gap between the navigation and positioning accuracy of AUVs compared to airborne and land-based ones, which needs to be eliminated [6]. This is mainly due to the fact that the Global Positioning System (GPS) is not available in the underwater environment [7].

AUV positioning using only onboard sensors such as a Doppler velocity logger (DVL) or an inertial measurement unit (IMU) will have an accumulated error [8]. AUVs are able to use their navigation sensors to obtain relevant measurement information for subsequent integration with underwater acoustic communication technologies, solving the positioning and navigation problem [9,10].

Due to the complexity of the marine environment, a single navigation method cannot meet the requirements of high accuracy and stability [11,12]. Approaches to determining the positioning error of AUVs are defined in [8]. These approaches are based on the fact that the AUV navigation complex consists of a hydroacoustic and onboard system. The onboard navigation system may include an inertial measurement unit (IMU), a Doppler velocity logger (DVL), and a pressure sensor (PS) [13]. Moreover, GPS, DVL, and PS separately provide position, velocity, and depth information, which are used to correct the navigation data [13–15].

In contrast to airborne or ground-based drones, AUVs are dealing with a uniquely difficult navigational problem due to the lack of high-precision satellite navigation underwater. Of course, for remotely piloted vehicles, additional navigation information (position, speed) may be sent to the vehicle via fiber optic cable. However, for unmanned submersibles without cable communication, this is almost impossible to implement in practice [16]. There are three main methods of AUV navigation in the literature: dead reckoning and inertial navigation, acoustic navigation, and geophysical navigation methods [17].

The first method is based primarily on inertial navigation equipment, which has become financially affordable, especially after the creation of microelectromechanical systems (MEMSs).

Since measurement errors of inertial navigation equipment are monotonically increasing and unlimited, other aids (e.g., differential global positioning system for position estimation, Doppler velocity log or correlated speed log for velocity estimation; pressure sensors for depth estimation, etc.) must be integrated to improve positioning system accuracy [18].

Acoustic navigation is based on using the AUV transponder's acoustic signals to determine its position. The most common methods are the long baseline, which uses at least two widely separated transponders mounted usually on the seafloor, and the ultra-short baseline, which uses GPS-calibrated transponders on an accompanying surface vessel. Both methods have a limited range (about 10 km for individual LBLs, about 4 km in deep water, and less than 0.5 km in shallow water for USBL networks). Because LBL requires the installation of beacons, its applicability is limited to missions performed in stationary locations (e.g., harbor defense). In addition, beacon installation and maintenance are complicated and expensive operations. USBL may not be applicable in some military applications because of tactical limitations, as it requires an accompanying vessel [19].

The most commonly used method of obtaining absolute position information underwater is through the buoys. These buoys are in known locations, and the AUV receives the range and/or azimuth to several of them and then calculates its position through trilateration or triangulation. Based on the location of the transceivers, three different basic systems can be distinguished: long baseline (LBL) systems, short baseline (SBL) systems, and ultra-short baseline (USBL) systems [20,21].

A typical configuration for a standard long baseline is shown in Figure 1a. Two or more buoys are deployed around the perimeter of the area in which the AUV will operate. These buoys are anchored and float on the surface or, especially in deeper waters, several meters above the seafloor. Each unit receives acoustic requested pings from a common receiving channel. After receiving a request ping from the AUV, each unit waits for a unique, specific response time and then sends a ping in response via its own separate transmission channel.

The standard LBL systems mentioned earlier are not well suited for large groups, as only one AUV can access the buoy network at a time and receive position updates. Therefore, the position update interval increases with the number of vehicles.

New LBL systems, like the one shown in Figure 1b, synchronized the clocks of the buoys and the AUV transceiver units. The buoys broadcast a ping containing a unique identifier at certain intervals. When the AUV receives this ping, the known beacon broadcast schedule and the timing of the synchronized clocks ensure that the vehicle knows when the ping was sent and can directly calculate the OWTT (one-way travel time).

Another improvement over conventional LBLs is the system shown in Figure 1c. Relying on the setup in Figure 1b, the buoys now transmit their GPS positions along with a unique identifier. As with the system described earlier, the AUV does not need to send queries to the buoys. With the buoy positions embedded in the ping, the buoys are free to swim, and there is no need to save their coordinates to the AUV before deployment.

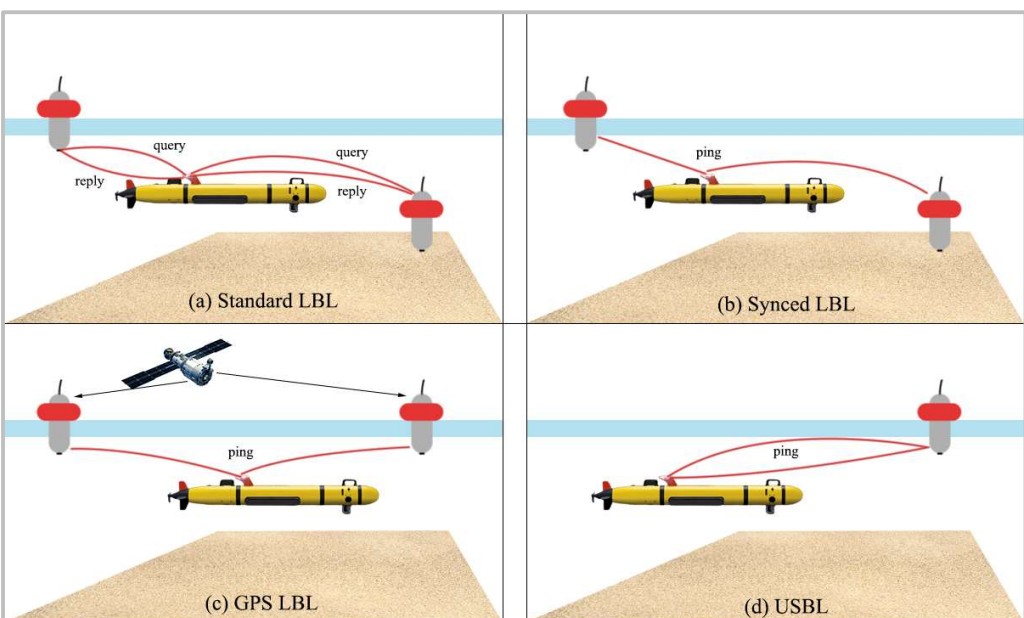

**Figure 1.** Buoy-based underwater positioning methods.

The beacons' placement depends on the task conditions. Two or more beacons are placed at known locations, either as buoys on the water surface or moored on the seafloor. To increase the area of AUV operation, an approach in which surface vehicles are moved by mobile beacons for AUVs is proposed in [22–24]. A similar approach is proposed in the "Widely scalable Mobile Underwater Sonar Technology" project [25].

Among the existing solutions for the positioning problem, the work [26] stands out, which considered a system of sensor data fusion, among which the values of slant range from three hydroacoustic buoys located on the surface. A similar problem was presented in [27], except that four buoys are already involved. In the presented sources, it is possible to allocate essential deficiencies, such as the absence of tuning parameters for the filter and sensor fusion methodology. Moreover, quality metrics for filter errors are not given, which makes it difficult to determine the accuracy of the state recovery algorithms.

The use of hydroacoustic communication devices to obtain relative observations is an effective and reliable measurement method. However, AUVs are used in complex marine environments where unfavorable conditions may affect the sensors, leading to unknown errors in the measurement system. This inevitably leads to a decrease in the accuracy and stability of the filtering procedure.

The main problem in solving the AUV navigation and positioning problem is the state estimation and accurate collection of observation information. Filtering algorithms for AUV state estimation are a key factor in guaranteeing satisfactory underwater navigation accuracy.

Various filter models and algorithms have been proposed to improve the accuracy of an integrated navigation system [7]. The Kalman filter (KF) is a well-known method for integrated navigation applications; see, for example, [11]. The traditional Kalman filter can be applied to linear systems. Most existing AUV positioning methods comprise improved Kalman filter (KF)-based navigation algorithms combined with measurement data from underwater navigation sensors. The model of a joint AUV positioning system is often nonlinear. Therefore, the extended Kalman filter (EKF) [28–30] and the uncentered Kalman filter (UKF) [31,32] are commonly used for state estimation. The accuracy of both EKF and UKF is affected by various factors, such as filter model and noise characteristics.

In this work, it will be acceptable to use two buoys on the surface to provide the positioning of the submersible when complexing information with inertial navigation

sensors (INSs/IMUs), a Doppler velocity logger (DVL), and a pressure sensor (PS). It is assumed that the buoys do not provide bearing angle data.

The hydroacoustic transceiver arrangement was based on [26], except for the existence of only two buoys. The studies [33,34] were used as a basis for mathematical modeling of the navigation system object dynamics (ANPA and buoys). Hydrodynamic parameters were determined using tabular data [35,36], and SDC-form [37] was used to represent all equations of dynamics and kinematics. The expressions presented in [38] were used to simulate the measurements of each sensor in the navigation system. The extended Kalman filter for state filtering was formed according to the general methodology presented in a number of sources [26–30]. The well-known maximum-ratio combining method was used as the considered method of sensor fusion.

Table 1 shows the overview of related work.

**Table 1.** Overview table.

| Reference | Key Features | Perspectives | Cons |
|---|---|---|---|
| [4] | A multilateration algorithm derived from Potluri's algorithm has been used to compute the receiver position | Fusion with additional sensors is in perspective | The system uses four buoys. No additional position sensors are used |
| [5] | A robust integrated navigation algorithm based on a maximum correntropy criterion and FGO scheme is proposed | - | The proposed method is the most computationally intensive |
| [26] | A data association layer and a dynamic SBL master selection heuristic were implemented | The extrinsic position of each beacon will be incorporated in the localization state vector, and thus, better results can be achieved | Using multiple transponders |
| [27] | An algorithm based on multi-beacon is proposed, and the propagation time is used as the measurement | - | The problem is considered in the 2D plane. The system uses four buoys. No additional position sensors are used |
| [28] | Proposed an improved Sage–Husa adaptive extended Kalman filter (improved SHAEKF) | Expanding from a two-dimensional plane to a three-dimensional space | The problem is considered in the 2D plane |
| [29] | The algorithm proposed in this paper adds the zero-space vector as the observability constraint, which improves the consistency of the cooperative positioning system | Improving the robustness of the system by reducing the communication frequency | No additional position sensors are used |

This paper has the following structure: Section 2 is devoted to the used methods and means for the positioning task, Section 3 describes the components of the proposed navigation system, and Section 4 is devoted to the simulation modeling. Section 5 contains conclusions on the work performed and further research directions.

## 2. Materials and Methods

### 2.1. Underwater Vehicle Model Description

General coordinates of an autonomous underwater vehicle (AUV) are determined in a geocentric coordinate system using SNAME notation [33,34]:

$$\eta = \begin{bmatrix} \eta_1 \\ \eta_2 \end{bmatrix} \in \mathbb{R}^6, \tag{1}$$

where $\eta_1 = \begin{bmatrix} x & y & z \end{bmatrix}^T$ determines the longitudinal, lateral, and vertical positions, respectively, and the vector $\eta_2 = \begin{bmatrix} \varphi & \theta & \psi \end{bmatrix}^T$ determines the Euler angles of roll, pitch, and yaw, respectively.

The velocity vector $v$ is expressed in the coordinate system associated with the body [33]. The velocities (linear and angular), according to the notation announced above, should be written as

$$v = \begin{bmatrix} v_1 \\ v_2 \end{bmatrix} = \begin{bmatrix} u & v & w & p & q & r \end{bmatrix}^T. \tag{2}$$

Under the influence of hydrodynamic effects caused by the aquatic environment, it is possible to write down the expression for the underwater vehicle dynamics in the following form:

$$\begin{aligned} M\dot{v} + C(v)v + D(v)v + g(\eta) &= \tau + J^T(\eta)\overline{f}_e, \\ M &= M_{RB} + M_A, \\ C(v) &= C_{RB}(v) + C_A(v), \end{aligned} \tag{3}$$

where $M_{RB} \in \mathbb{R}^{6\times6}$—solid body inertia matrix (6 is the DoF number),
$M_A \in \mathbb{R}^{6\times6}$—added mass matrix,
$C_{RB}(v) \in \mathbb{R}^{6\times6}$—solid body Coriolis matrix,
$C_A(v) \in \mathbb{R}^{6\times6}$—added mass Coriolis matrix,
$D(v) \in \mathbb{R}^{6\times6}$—dissipative coefficients matrix,
$g(\eta) \in \mathbb{R}^{6\times1}$—gravitational forces and moments vector,
$\tau \in \mathbb{R}^{6\times1}$—vector (of forces and moments) of the controls applied to the body expressed in the body-fixed frame,
$\overline{f}_e \in \mathbb{R}^{6\times1}$—vector (of forces and moments) of external disturbances applied to the body expressed in the inertial frame.

In real systems, $\tau$ is calculated with the thruster's models using the thruster's distribution matrix ($T$) as

$$\tau = Tu, \tag{4}$$

where $u \in \mathbb{R}^{p\times1}$ is a vector that describes the thrusters' load ($p$ is the number of thrusters), and the matrix $T \in \mathbb{R}^{n\times p}$ relates the thrusters' load $u$ and the forces/moments vector $\tau$.

The kinematics equation linking (1) and (2) is written in the form

$$\dot{\eta} = J^{-1}(\eta)v = \begin{bmatrix} R_I^B(\eta) & 0_{3\times3} \\ 0_{3\times3} & J_{k,o}(\eta) \end{bmatrix}^{-1} v = \begin{bmatrix} R_B^I(\eta) & 0_{3\times3} \\ 0_{3\times3} & J_{k,o}^{-1}(\eta) \end{bmatrix} v, \tag{5}$$

where $R_I^B(\eta) \in \mathbb{R}^{3\times3}$ is the rotation matrix obtained from the Euler angles [33], which links the navigation (inertial) frame and the underwater vehicle (body) frame, and $J_{k,o}(\eta) \in \mathbb{R}^{3\times3}$ is the Jacobian linking the angular velocities of the world reference system and the body system [33].

Given $x = \begin{bmatrix} \eta \\ \dot{\eta} \end{bmatrix}^T$, it is possible to write the state vector

$$\dot{x} = \begin{bmatrix} \dot{\eta} \\ M^{-1}\left(Tu + J^T(\eta)\overline{f}_e - C(v)v - D(v)v - g(\eta)\right) \end{bmatrix}. \tag{6}$$

The latter equation can be represented in the state-dependent coefficients (SDCs) form [37]

$$\frac{dx}{dt} = A(x,t)x + B(x,t)\overline{u}, \ \ x(t_0) = x_0,$$

$$y = C(t)x + \mathcal{D}(t)\overline{u},$$

$$A(x,t)x = \begin{bmatrix} 0_{6\times6} & J(x)^{-1} \\ 0_{6\times6} & -M^{-1}(C(x)+D(x)) \end{bmatrix}, \ \ B(x,t) = \begin{bmatrix} 0_{6\times6} \\ M^{-1} \end{bmatrix},$$

$$\overline{u} = Tu + u_{add}(x), u_{add} = J^T(x)\overline{f}_e - g(x).$$

(7)

where $A \in \mathbb{R}^{2n\times2n}$—dynamics matrix,
$B \in \mathbb{R}^{2n\times n}$—control matrix,
$C \in \mathbb{R}^{m\times2n}$—output matrix ($m$—number of system outputs),
$\mathcal{D} \in \mathbb{R}^{m\times n}$—input–output coupling matrix,
$\overline{u} \in \mathbb{R}^{n\times1}$—control vector, including the vector of external perturbations $u_{add} \in \mathbb{R}^{n\times1}$.

### 2.2. Description of the Controlled Object MMT-300

The MMT-300 AUV is designed to perform bottom and aquatic surveys at depths of up to 300 m. Search programs (missions) can be described as AUV tacking through the surveyed area with activation of onboard search devices (one or several) at specified intervals and then returning the vehicle to the supplying vessel. The visual appearance of the AUV MMT-300 is shown in Figure 2.

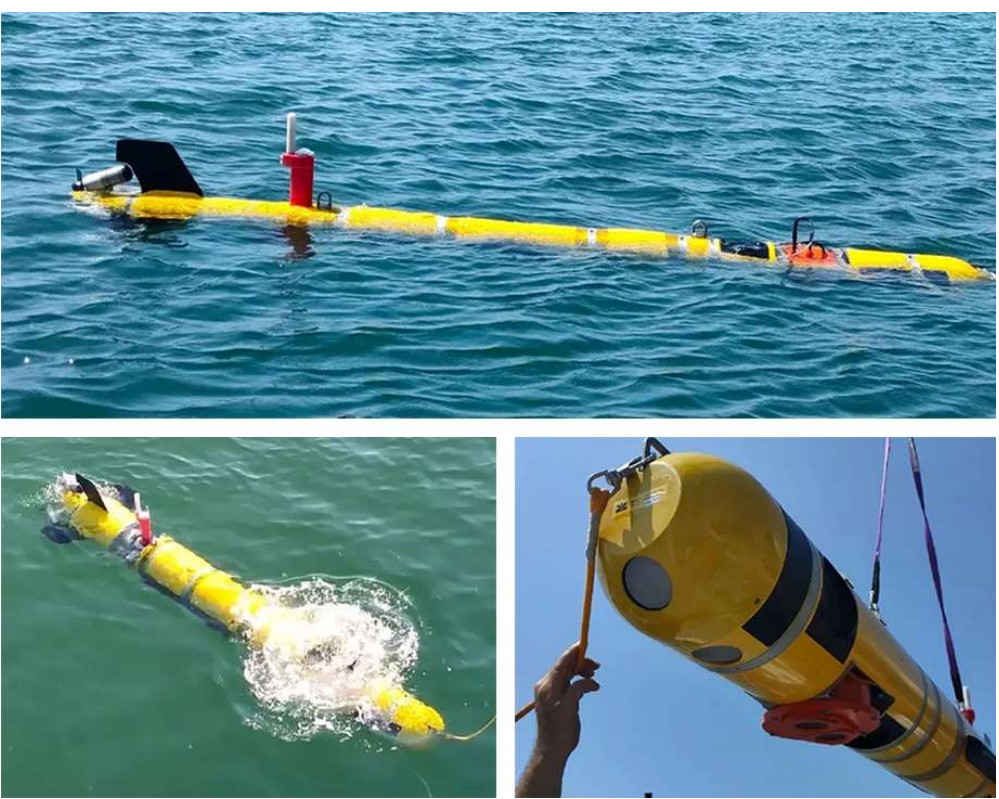

**Figure 2.** The visual appearance of the AUV MMT-300.

The locations of the main AUV elements are shown in Figure 3.

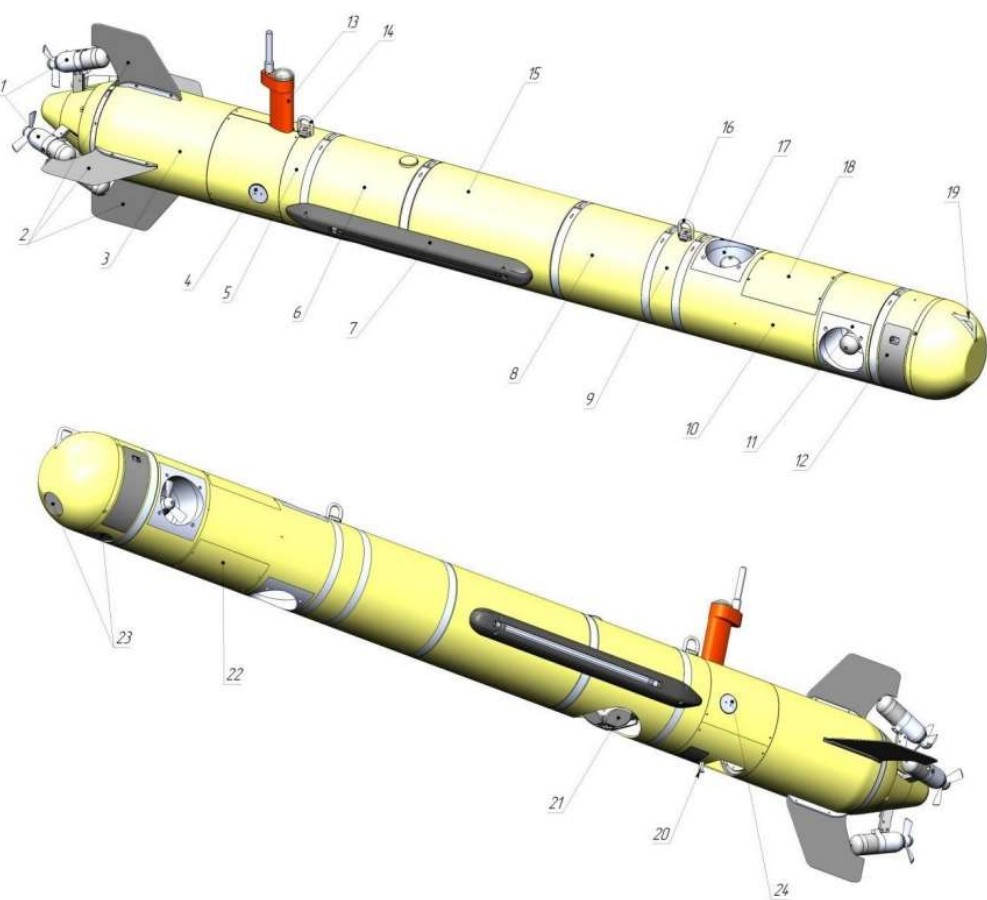

**Figure 3.** Location of the AUV's main components: 1—sustainer propulsion system, 2—stabilizers, 3—aft compartment, 4—charge connector, 5—aft cover, 6—navigation and communication compartment, 7—side-scan sonar, 8—hermetic payload compartment, 9—bow cover, 10—outboard payload compartment, 11—horizontal thruster, 12—compartment cover, 13—radio module, 14, 16—cargo arm, 15—autopilot compartment, 17—vertical thruster, 18, 22—payload compartment cover, 19—towing arm, 20—emergency ballast mechanism, 21—Doppler velocity logger, 23—ELS antennas, 24—hermetic communication connector.

### 2.2.1. Software Control and Onboard Navigation System

The onboard control and navigation system (OCNS) is designed to control all systems of the vehicle in all operating modes of the AUV.

The OCNS provides the following:

- Execution of a pre-programmed AUV task;
- Detection of emergencies and their adequate handling;
- Trajectory control of various types;
- Determination of the resulting AUV location in geographical coordinates.

The onboard control and navigation system includes the following:

- Emergency sensors and actuators (water sensors, voltage sensor, power failure and low battery detectors, thrusters);
- Magnetic compass and orientation sensors;
- Radio module;
- Depth sensor;
- Doppler velocity logger.

### 2.2.2. Propulsion–Steering Complex

The propulsion–steering complex (PSC) is a program-controlled executive device. The AUV uses a propulsion system consisting of four aft reversible thrusters and two thrusters of horizontal and vertical channels. Stern thrusters are located in pairs in the horizontal and vertical planes at an angle of 22° to the longitudinal axis. This propulsion scheme makes it possible to create arbitrary forces and moments to control the AUV, as well as to implement various movement modes. The AUV motion is controlled by five degrees of freedom. The range of possible values for a given longitudinal velocity is 0–2.0 m/s.

### 2.2.3. Hydroacoustic Navigation and Communication System (Option)

The HNCS provides vehicle tracking, as well as periodic correction in determining the onboard coordinates of the AUV. The error in determining the distance depends significantly on the hydrological conditions of the working area, as well as on the error in the set sound velocity. In the absence of HANS, the AUV uses calculus (using magnetic compass (MC), depth sensor (DS), and Doppler velocity logger (DVL) readings) and GNSS (on the surface) to determine its own coordinates.

The main HNCS features are summarized in Table 2.

**Table 2.** HNCS features.

| Parameter | Value |
|---|---|
| Operating depth, m | up to 300 |
| Operating range (slant), km | up to 3.5 |
| Slant-range measurement error, m | no more than 0.01 |
| Operating frequency, kHz | 18 . . . 34 |
| Bearing measurement error, deg | 0.1 |
| Data transfer rate, kbit/s | up to 13.9 |

### 2.3. MMT-300 Mathematical Model

The main MMT-300 vehicle parameters are summarized in Table 3.

**Table 3.** AUV parameters.

| Mass, $m$ | Length, $L$ | Radius, $r$ | Center-of-Mass Vector, $r_g^b$ | Buoyancy Center Vector, $r_b^b$ | Buoyancy, $B$ |
|---|---|---|---|---|---|
| 150 kg | 3.081 m | 0.147 m | $\begin{bmatrix} 0 & 0 & 0 \end{bmatrix}^T$ m | $\begin{bmatrix} 0 & 0 & 0.07 \end{bmatrix}^T$ m | mg, N |

### 2.3.1. Thrusters Allocation Matrix

Given the geometric and physical parameters of the thrusters, it is possible to determine the thrusters' allocation matrix (4) by the following expression

$$t_i = \begin{bmatrix} f_i \\ r_i \times f_i \end{bmatrix}, \quad T = \begin{bmatrix} t_1 & \cdots & t_n \end{bmatrix}, \tag{8}$$

where $f_i$ is the force created by the $i$-th thruster, and $r_i$ is the geometric position of the force application point of the $i$-th thruster relative to the vehicle's center of mass.

It is worth noting that the forces generated by the MMT-300 thrusters depend on the software control, which leads expression (8) to the parametric form

$$t_i(code) = \begin{bmatrix} f_i(code) \\ r_i \times f_i(code) \end{bmatrix}, \quad T(code) = \begin{bmatrix} t_1(code) & \cdots & t_p(code) \end{bmatrix}, \tag{9}$$

where $code \in [-128; 128]$ is the software control code.

From Figure 4, it is possible to determine the values $r_i$, which are presented in Table 4.

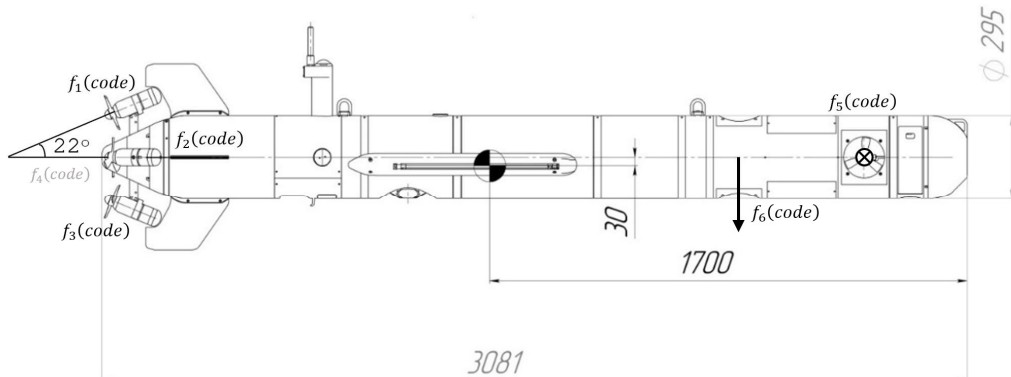

**Figure 4.** MMT-300 geometric parameters.

**Table 4.** Thrusters' moments of arms of the MMT-300.

| Thruster No. | $r_i$, (mm) | | |
|:---:|:---:|:---:|:---:|
| | $l_{x_i}$, (mm) | $l_{y_i}$, (mm) | $l_{z_i}$, (mm) |
| 1 | $-1330$ | 0 | 189 |
| 2 | $-1330$ | 159 | 30 |
| 3 | $-1330$ | 0 | $-129$ |
| 4 | $-1330$ | $-159$ | 30 |
| 5 | 1330 | 0 | 30 |
| 6 | 882 | 0 | 30 |

Since all thrusters are the same in design, they create a single force that depends on the software control code $F(code)$. In this case, $f_i(code)$ is defined by taking into account the geometry of the control object as

$$f_1(code) = \begin{bmatrix} F(code)\cos(22°) \\ 0 \\ F(code)\sin(22°) \end{bmatrix}, \ f_2(code) = \begin{bmatrix} F(code)\cos(22°) \\ F(code)\sin(22°) \\ 0 \end{bmatrix},$$

$$f_3(code) = \begin{bmatrix} F(code)\cos(22°) \\ 0 \\ -F(code)\sin(22°) \end{bmatrix}, \ f_4(code) = \begin{bmatrix} F(code)\cos(22°) \\ -F(code)\sin(22°) \\ 0 \end{bmatrix}, \quad (10)$$

$$f_5(code) = \begin{bmatrix} 0 \\ F(code) \\ 0 \end{bmatrix}, \ f_6(code) = \begin{bmatrix} 0 \\ 0 \\ -F(code) \end{bmatrix}.$$

From the thruster's static characteristic (Figure 5), it is possible to write down the functional dependence between the control code and the generated force, for example, by means of a sixth-order polynomial

$$F(code) = 7.2937 \cdot 10^{-13} \cdot code^5 - 6.9155 \cdot 10^{-10} \cdot code^4 - 2.2935 \cdot 10^{-8} \cdot code^3 + \\ 2.3559 \cdot 10^{-5} \cdot code^2 + 1.9049 \cdot 10^{-4} \cdot code + 0.1332. \quad (11)$$

With the dependencies defined above, it is possible to determine $T(code)$, which makes it possible to define the expression for forces and moments (4) already in parametric form $\tau(code) = T(code) \cdot u$. In this case, $u$ determines the thruster's load percentage. At full performance, $u = \begin{bmatrix} 1 & 1 & 1 & 1 & 1 & 1 \end{bmatrix}^T$.

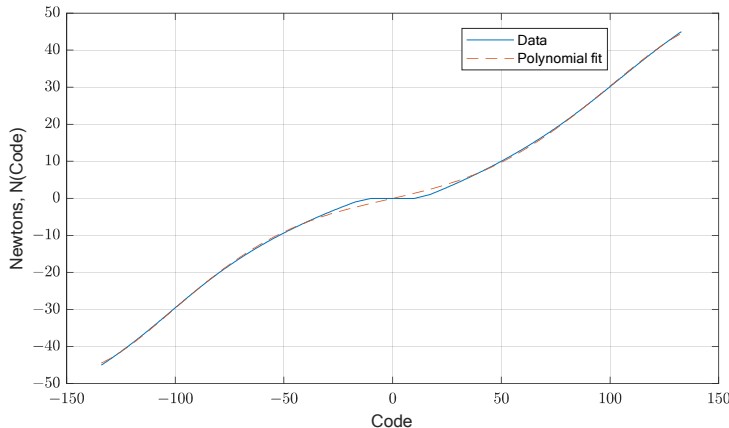

**Figure 5.** Static characteristics of the MMT-300 thrusters.

### 2.3.2. Inertia and Added Masses Matrices

The matrix $M_{RB}$ in (3) is constant, symmetric, and positively determined ($M_{RB} > 0$). The values filling it largely depend on the geometric configuration and, in the most general case, have the following form [33,34]

$$M_{RB} = \begin{bmatrix} mI_{3\times3} & -mS\left(r_g^b\right) \\ mS\left(r_g^b\right) & I_{O_b} \end{bmatrix} \in \mathbb{R}^{6\times6}, \tag{12}$$

where $m$—rigid body mass,
$I_{3\times3}$—$3 \times 3$ identity matrix,
$I_{O_b}$—inertia tensor in the reference frame of the body,
$r_g^b = \begin{bmatrix} x_g & y_g & z_g \end{bmatrix}^T$—vector from the origin to the gravity center of the body,
$S(\,\cdot\,)$—operator of vector transformation into a skewed matrix [33].

The inertia tensor for a solid cylinder is defined as

$$I_{O_b} = \begin{bmatrix} \dfrac{1}{2}mr^2 & 0 & 0 \\ 0 & \dfrac{1}{12}m\left(3r^2 + L^2\right) & 0 \\ 0 & 0 & \dfrac{1}{12}m\left(3r^2 + L^2\right) \end{bmatrix}. \tag{13}$$

Given the symmetric form approximation of the underwater vehicle, the matrix notation will be greatly simplified and will take a diagonal form [33]:

$$M_A = -diag\left\{X_{\dot{u}},\ Y_{\dot{v}},\ Z_{\dot{w}},\ K_{\dot{p}},\ M_{\dot{q}},\ N_{\dot{r}}\right\} = \begin{bmatrix} M_A^{Tr} \in \mathbb{R}^{3\times3} & \ddots \\ \ddots & M_A^{Rot} \in \mathbb{R}^{3\times3} \end{bmatrix} \tag{14}$$

There is no uniquely correct way to calculate the $M_A$ matrix elements, so, as a rule, a variety of estimation methods are used. For a solid cylinder of mass $m$, length $L$, and radius $r$, the added masses (14) are defined as [33]

$$\begin{aligned} X_{\dot{u}} &= -0.1m, \\ Y_{\dot{v}} &= -\pi\rho r^2 L, \\ Z_{\dot{w}} &= -\pi\rho r^2 L, \\ K_{\dot{p}} &= 0, \\ M_{\dot{q}} &= -\frac{1}{12}\pi\rho r^2 L^3, \\ N_{\dot{r}} &= -\frac{1}{12}\pi\rho r^2 L^3. \end{aligned} \tag{15}$$

### 2.3.3. Coriolis Matrix

According to Fossen [34], the matrix $C(v)$ is generally defined from the blocks of the inertia matrix $M$. In this case, $M = \begin{bmatrix} M_{11} & M_{12} \\ M_{21} & M_{22} \end{bmatrix}$, then

$$C(v) = \begin{bmatrix} 0_{3\times 3} & -S(M_{11}v_1 + M_{12}v_2) \\ -S(M_{11}v_1 + M_{12}v_2) & -S(M_{21}v_1 + M_{22}v_2) \end{bmatrix} \in \mathbb{R}^{6\times 6}. \tag{16}$$

### 2.3.4. Gravitational and Buoyancy Vector

In [34], to calculate $g(\eta)$, the author uses the expression

$$g(\eta) = \begin{bmatrix} (mg - B)sin\theta \\ -(mg - B)cos\theta sin\phi \\ -(mg - B)cos\theta cos\phi \\ -(y_g mg - y_b B)cos\theta cos\phi + (z_g mg - z_b B)cos\theta sin\phi \\ (z_g mg - z_b B)sin\theta + (x_g mg - x_b B)cos\theta cos\phi \\ -(x_g mg - x_b B)cos\theta sin\phi - (y_g mg - y_b B)sin\theta \end{bmatrix}, \tag{17}$$

where $m$—vessel's mass including water in space,
g—acceleration of gravity,
$B = \rho g \nabla$—buoyancy force,
where $\rho$—liquid density,
$\nabla$—liquid volume displaced by the vessel,
$x_b$, $y_b$, $z_b$—components of the vector $r_b^b = \begin{bmatrix} x_b & y_b & z_b \end{bmatrix}^T$ from the origin to the center of buoyancy.

### 2.3.5. Damping Forces and Moments

The drag force/lifting force (Figure 6) is calculated with

$$F_D = C_D \left(\frac{\rho v^2}{2}\right) S_D, \ \ F_L = C_L \left(\frac{\rho v^2}{2}\right) S_L, \tag{18}$$

where $C_D$—drag coefficient,
$C_L$—lift coefficient,
$\rho$—environmental density,
$v$—body velocity in the fluid,
$S_D$—surface area, relative to the flow,
$S_L$—surface area perpendicular to the flow.

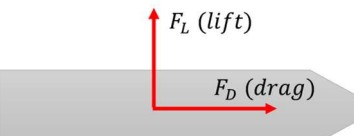

**Figure 6.** Damping force distribution diagram.

The $C_L$ coefficients for the cylindrical approximation form AUV are given in [35]. For a given ratio of length and radius, the corresponding coefficient is $C_L = 0.64$.

The $C_D$ coefficient is determined relative to the bow cover's geometry. In this case, the closest shape is a hemispherical bowl. In [36], the corresponding value of $C_D = 0.4$.

The frontal and side areas are determined solely from the basic geometric parameters of the submersible by the following expressions:

$$S_D = \pi(0.147)^2 = 0.06789 \ (\text{m}^2).$$
$$S_L = 2\pi \, 0.147 \, 3.081 = 2.8457 \ (\text{m}^2). \tag{19}$$

The forces induced by the cylindrical body motion, $F_{DAMP} = \begin{bmatrix} F_D^{OX} & F_L^{OY} & F_L^{OZ} \end{bmatrix}^T$, are approximately defined as

$$F_D^{OX} = 0.4 \left( \tfrac{1000 \cdot u^2}{2} \right) 0.06789 = 13.578 u^2,$$
$$F_L^{OY} = 0.64 \left( \tfrac{1000 \cdot v^2}{2} \right) 2.8457 = 910.624 v^2, \tag{20}$$
$$F_L^{OZ} = 0.64 \left( \tfrac{1000 \cdot w^2}{2} \right) 2.8457 = 910.624 w^2,$$

The moments created by the cylindrical body motion are approximately defined as

$$M_{DAMP} = v_1 \times M_A^{Tr} v_1. \tag{21}$$

The damping forces, taking into account the calculated parameters, are defined as

$$D(v)v = diag\{F_{DAMP}, M_{DAMP}\}. \tag{22}$$

### 2.4. Mathematical Model of the Buoy

The scheme of the buoys involved in the system is shown in Figure 7. Their parameters are summarized in Table 5.

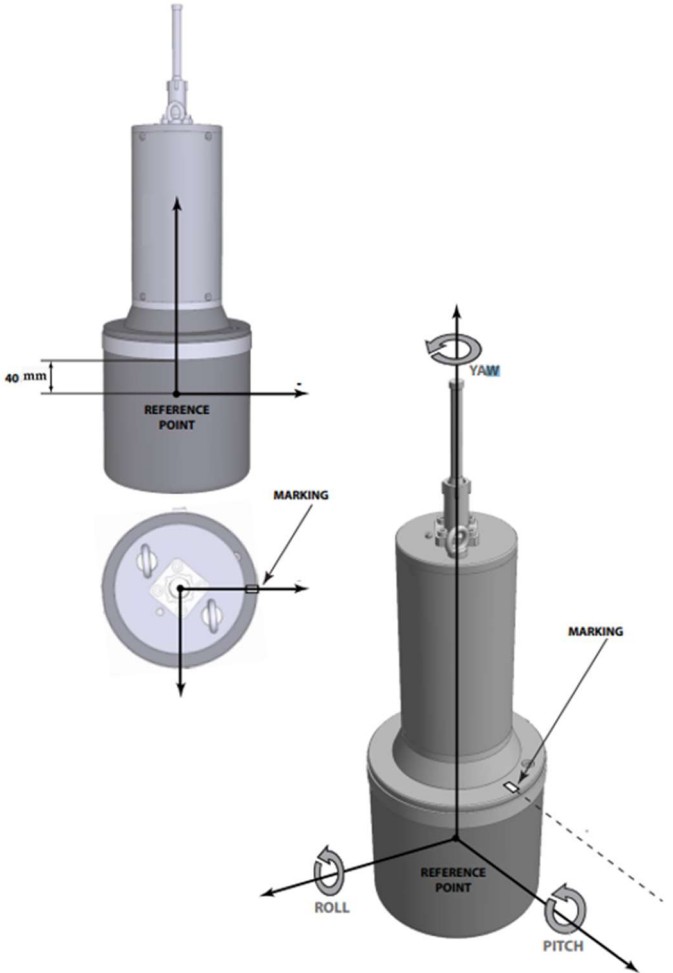

**Figure 7.** The schematic representation of the buoy in its reference frame.

**Table 5.** Buoy parameters.

| Mass, $m$ | Length, $L$ | Radius, $r$ | Center-of-Mass Vector, $r_g^b$ | Buoyancy Center Vector, $r_b^b$ | Buoyancy, $B$ |
|---|---|---|---|---|---|
| 12 kg | 0.36 m | 0.08 m | $\begin{bmatrix} 0 & 0 & -0.16 \end{bmatrix}^T$ m | $\begin{bmatrix} 0 & 0 & 0.21 \end{bmatrix}^T$ m | 171, N |

Given that the buoy has a cylindrical shape, it is possible to describe its dynamics similar to MMT-300, except that the inertia tensor is rotated and defined as

$$
I_{O_b} = \begin{bmatrix} \frac{1}{12}m(3r^2 + L^2) & 0 & 0 \\ 0 & \frac{1}{12}m(3r^2 + L^2) & 0 \\ 0 & 0 & \frac{1}{2}mr^2 \end{bmatrix}.
\tag{23}
$$

*2.5. Description of Onboard Sensors*

No one sensor can measure perfectly, i.e., without errors. An error is the difference between the true value and the actual measured value. Sensor errors are divided into random and systematic components. Random components are also called random measurement errors. Their specific realization at a particular time cannot be predicted; it is only possible to describe a general pattern of their behavior. The mathematical model of random components is defined as a sequence of the white noise counts sum with Markov process counts or flicker noise [38]. Determining the characteristics of the random processes allows us to correctly describe their dynamics when integrating the IMU with other sensors, improving the filtering quality.

2.5.1. Inertial Sensors

When combining the IMU with other sensors, the zero-point biases of accelerometers and gyroscopes are included in the vector of estimated parameters; this allows us to take into account their random component, which is the main source of positioning errors after compensating for other systematic errors. The temperature has the greatest effect on the zero-offset of MEMS sensors; at the same time, its influence on the axis unorthogonality and scale coefficient errors is limited.

The output vector of velocity measurements from the 3-axis accelerometer unit is modeled as

$$
\tilde{v}_{IMU} = v + w_{IMU} + b_{IMU},
\tag{24}
$$

where $w_{IMU} \sim \mathcal{N}(0, \sigma_{IMU}^2)$, $\sigma_{IMU}^2$ is the RMS error of the accelerometer, and $b_{IMU}$ is the acceleration bias, which is modeled as a 1st-order Markov process.

Similarly, the real output of the gyroscope is defined as

$$
\tilde{\eta}_{2GYRO} = \eta_2 + w_{GYRO} + b_{GYRO},
\tag{25}
$$

where $w_{GYRO} \sim \mathcal{N}(0, \sigma_{GYRO}^2)$, $\sigma_{GYRO}^2$ is the RMS error of the accelerometer, and $b_{GYRO}$ is the gyroscope bias, which is modeled as a 1st-order Markov process.

2.5.2. Depth Sensors

Depth and underwater pressure have a direct correlation. As the unit goes deeper into the water, the pressure readings increase linearly.

The actual depth sensor output is simulated by adding noise to the actual depth:

$$
\tilde{z} = z + w_z,
\tag{26}
$$

where $w_z \sim \mathcal{N}(0, \sigma_z^2)$ and $\sigma_z^2$ is the RMS error in depth.

### 2.5.3. Doppler Velocity Logger

The Doppler logger measures the change in acoustic frequency to determine the vehicle's speed relative to the seafloor. The actual Doppler logger output is modeled by adding noise to the actual vehicle speed:

$$\begin{bmatrix} \widetilde{u}_{DVL} \\ \widetilde{v}_{DVL} \end{bmatrix} = \begin{bmatrix} u \\ v \end{bmatrix} + \begin{bmatrix} w_u \\ w_v \end{bmatrix}, \tag{27}$$

where $w_u \sim \mathcal{N}(0, \sigma_{DVL}^2)$, $w_v \sim \mathcal{N}(0, \sigma_{DVL}^2)$, and $\sigma_{DVL}^2$ is the RMS of the velocity error.

### 2.5.4. Underwater Acoustic Positing System

The underwater acoustic positing system measures the distance and direction of the vehicle from the reference positions. It can be interfaced with GPS to provide Earth-related coordinates. However, acoustic position estimate is affected by GPS accuracy, system installation, ship attitude, sound velocity profile, ray bending, and measurement noise [18].

Assuming the system is precisely calibrated, buoy installation and attitude have negligible effects. The mathematical model of the system's actual output is given as

$$\widetilde{p}_{UAPS} = p_{UAPS} + w_{UAPS} + b_{UAPS}, \tag{28}$$

where $w_{UAPS} \sim \mathcal{N}(0, \sigma_{UAPS}^2)$, $\sigma_{UAPS}^2$ is the RMS positional error, and $b_{UAPS}$ is the time-varying bias modeled as a 1st-order Markov process and depends on the sound velocity profile and ray-bending effect.

## 3. Navigation System Components

### 3.1. AUV Navigation System

In this work, the USBL positioning method is taken into consideration. The principle of USBL and LBL operation is based on the range measurements to the responding beacons with known coordinates. The onboard AUV transmitter emits a hydroacoustic signal, and the receiver records the response signal from the beacon. The range is determined by measuring the propagation time of the hydroacoustic signal. The error in determining the location of the vehicle depends on the accuracy of setting the beacon coordinates, determining the effective sound speed, and the accuracy of fixing the moments of the arrival response from the beacon.

USBLs use a beacon that emits hydroacoustic pulses without prior interrogation. The information delivered by the beacon and the submersible must be synchronized. The measured quantity is the time of signal propagation, as well as the phase difference of the signal's arrival at the receiving antenna elements mounted on the AUV. The used USBL system parameters are shown below:

- Operating range, <2000 m;
- Slant-range accuracy, 0.01 m;
- Acoustic connection, up to 31.2 kbit/s;
- Update rate, ≤1 Hz.

The specifications of the operator's computer with position correction software are as follows:

- Intel Core i5 11th Gen 1155G7 (2.50 GHz);
- 8 GB memory 512 GB NVMe SSD;
- Intel Iris Xe graphics.

As can be seen in Figure 8, it is necessary to provide measurements fusion of different sensors in order to estimate AUV position together with errors.

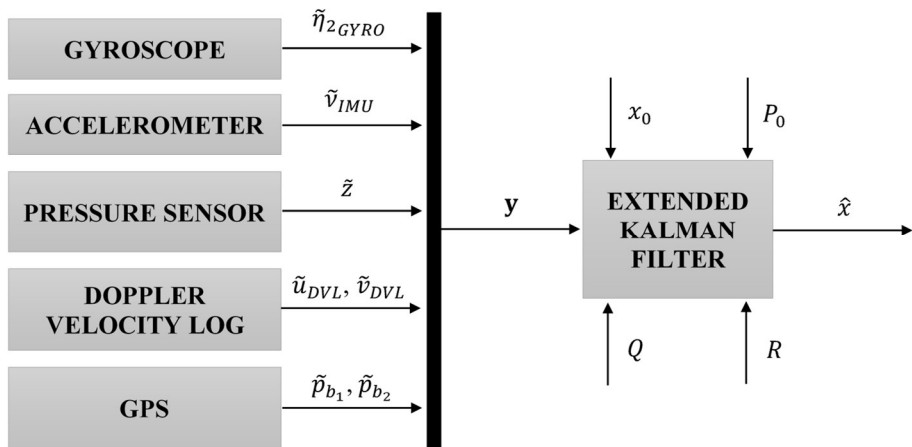

**Figure 8.** Fusion and filtering scheme for sensor readings.

*3.2. Slant Range Usage*

If there are two or more buoys, it is possible to reconstruct information about the location of the modem object using the slant-range mechanism to recalculate the position in the global coordinate system by GPS or GLONASS.

Two buoys are considered for the task. Their locations at longitudinal, lateral, and vertical displacements are determined by the $\eta_1^{(b_1)} = \begin{bmatrix} x_{b_1} & y_{b_1} & z_{b_1} \end{bmatrix}^T$ and $\eta_1^{(b_2)} = \begin{bmatrix} x_{b_2} & y_{b_2} & z_{b_2} \end{bmatrix}^T$ vectors. Assuming that the depth is constant (without considering wave perturbations), it is possible to neglect the $z_{b_1}$ and $z_{b_2}$ values, obtaining the positions in the planes $p_{b_1} = \begin{bmatrix} x_{b_1} & y_{b_1} \end{bmatrix}^T$ and $p_{b_2} = \begin{bmatrix} x_{b_2} & y_{b_2} \end{bmatrix}^T$.

The possibility of obtaining the slant range to the underwater vehicle will allow us to calculate its position but preliminarily requires knowledge of its depth $z$ (assuming that the pressure sensor is always working) to determine the projection of the range on the XY plane. Denoting the slanted ranges as $r_{b_1}$ and $r_{b_2}$, expressions for their plane projections are obtained as follows:

$$\begin{aligned} \bar{r}_{b_1} &= \sqrt{r_{b_1}^2 - z^2}, \\ \bar{r}_{b_2} &= \sqrt{r_{b_2}^2 - z^2}. \end{aligned} \tag{29}$$

Further, the positioning problem is reduced to the geometric problem of finding intersection points between two circles whose centers are the buoy positions and whose radii are the projections found earlier ($\bar{r}_{b_1}$ and $\bar{r}_{b_2}$).

The distance between the circles $d = |p_{b_2} - p_{b_1}|$ allows the use of the cosine theorem:

$$\cos \alpha = \frac{\bar{r}_{b_1}^2 + d^2 - \bar{r}_{b_2}^2}{2d\bar{r}_{b_1}}. \tag{30}$$

The possible positions of the AUV are now defined by the expression

$$\begin{bmatrix} x_{SR} \\ y_{SR} \end{bmatrix} = \bar{r}_{b_1} + \vec{U}_d \left( \bar{r}_{b_1} \cos \alpha \right) \pm \vec{U}_\perp \left( \bar{r}_{b_1} \sqrt{1 - \cos^2 \alpha} \right), \tag{31}$$

where $\vec{U}_d = \dfrac{p_{b_2} - p_{b_1}}{d} = \begin{bmatrix} \vec{U}_{d_x} \\ \vec{U}_{d_y} \end{bmatrix}$ is the unit vector between the first and the second center,

and $\vec{U}_\perp = \begin{bmatrix} \vec{U}_y \\ -\vec{U}_x \end{bmatrix}$ is perpendicular to the unit vector.

The duality of solving the equation is resolved if the previous underwater vehicle position is known. In this case, it is necessary to choose the point closest to the last vehicle position (Figure 9).

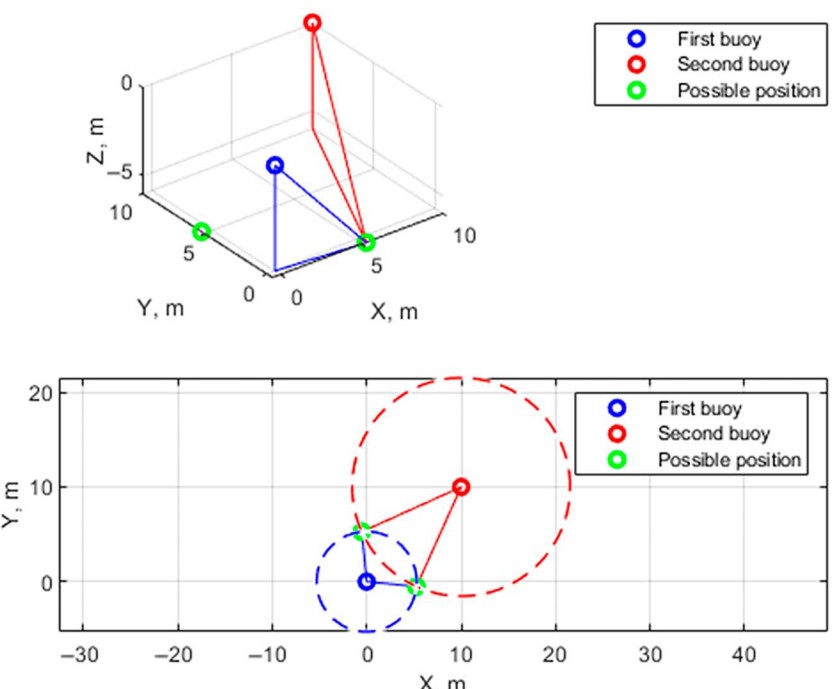

**Figure 9.** Slant ranges geometries.

### 3.3. Description of the Kalman Filter Model

Consider the problem of constructing a filter for a nonlinear system, known as an extended Kalman filter. It allows us to estimate the controlled object state, even if the dimensionality of the state vector of the system under study exceeds the number of measured parameters. To calculate the current state of an a priori known dynamic system, it is necessary to know its current measured state, as well as the state of the filter at the time of the previous measurement. The purpose of the filter is to minimize the sum of squared errors of the state vector estimation [28–30].

Taking into account the white noise of the system, $w$, and the white noise of measurements, $v$, the system in SDC form has the following view:

$$
\begin{aligned}
\frac{dx}{dt} &= A(x,t)x + B(x,t)\overline{u} + Gw, \quad x(t_0) = x_0, \\
y &= C(t)x + \mathcal{D}(t)\overline{u} + Hw + v,
\end{aligned}
\tag{32}
$$

The state vector estimation of the Kalman filter is defined as

$$
\begin{aligned}
\frac{d\hat{x}}{dt} &= A(x,t)\hat{x} + B(x,t)\overline{u} + K_{KF}(t)[y - C(t)\hat{x}], \quad \hat{x}(t_0) = \hat{x}_0, \\
K_{KF}(t) &= P(t)C(t)^T R^{-1}, \quad P(t_0) = P_0, \\
\dot{P}(t) &= A(x,t)P(t) + P(t)A(x,t)^T + GQG^T - P(t)C(t)^T R^{-1}C(t)P(t),
\end{aligned}
\tag{33}
$$

where $P(t)$ is the state covariance estimation, $P_0$ is the initial value of the state covariance estimation, $K_{KF}(t)$ is the Kalman filter gain matrix, and $Q$ and $R$ are the cost matrices in the Riccati equation.

Noises $w$ and $v$ must satisfy the following conditions:

$$
E(w) = E(v) = 0, \quad E\left(ww^T\right) = Q, \quad E\left(vv^T\right) = R.
\tag{34}
$$

### 3.4. Sensor Fusion

According to Figure 8, it is required to combine information from the sensors to form a measurement vector for subsequent filtering. There are numerous methods of combining sensor readings.

In telecommunications, maximum-ratio combining (MRC) is a data fusion method in which the signals of each channel are summed, and the gain of each channel is set proportional to the RMS signal level and inversely proportional to the RMS noise level in that channel:

$$s_{opt} = \frac{\sigma_{s_y}^2 s_x + \sigma_{s_x}^2 s_y}{\sigma_{s_x}^2 + \sigma_{s_y}^2}. \tag{35}$$

Maximum-ratio combining is the optimal fusion tool for independent channels with additive white Gaussian noise. If hydroacoustic buoys unambiguously determine the position, it is necessary to combine the Doppler logger data with the IMU in the measurements vector:

$$\widetilde{u} = \frac{\sigma_{DVL}^2 \widetilde{u}_{IMU} + \sigma_{IMU}^2 \widetilde{u}_{DVL}}{\sigma_{IMU}^2 + \sigma_{DVL}^2}, \widetilde{v} = \frac{\sigma_{DVL}^2 \widetilde{v}_{IMU} + \sigma_{IMU}^2 \widetilde{v}_{DVL}}{\sigma_{IMU}^2 + \sigma_{DVL}^2}. \tag{36}$$

## 4. Navigation System Modeling

The filtering software simulation considers the following situation: the MMT-300 moves to a predetermined point, and two hydroacoustic buoys are located on the water surface. The influence of surface current is taken into account by influencing the displacement of buoys. If the measurement error is larger than the state uncertainty estimate, the filter will "trust" the simulation data more. That is why it is important to correctly select the values of covariance matrices, the main tool for tuning the filter. With the given measurement devices' noise parameters, the EKF parameters are as follows:

$$\begin{aligned} P_0 &= I_{12\times12}, Q = 0, \\ \hat{x}_0 &= x_0 + w_{x_0} \sim \mathcal{N}(0, 0.05), \\ R &= diag\{0.8; 0.8; 0.102; 0.15; 0.15; 0.15; 0.25; 0.25; 0.35; 0.14; 0.14; 0.14\}. \end{aligned} \tag{37}$$

It is assumed that a Doppler logger is used when fusing sensor information due to the small distance from the seafloor. It should also be noted that zero-order extrapolators with update frequencies, presented in the Table 6, are used to simulate the discrete nature of the measurement devices.

**Table 6.** Update rates of the sensors.

| Sensor | Update Rate |
|---|---|
| IMU | 50 Hz |
| GPS | 1 Hz |
| DVL | 25 Hz |
| PS | 50 Hz |

The block diagram of the system model is shown in Figure 10 and reflects its main components: hydroacoustic buoys to obtain slant ranges to AUV, a controller to send control signals, the dynamics/kinematics block of the underwater vehicle, and the sensor fusion block.

The simulation results are presented below. The initial state of the underwater vehicle, in this case, is defined as zero. Figure 11 shows the trajectories of the submersible and the hydroacoustic buoys.

The charts of the system output are shown in Figure 12. The measurement sensor errors are evident here: noise and high update frequencies are noticeable. The results of the Kalman filter for state recovery are shown in Figure 13 and show more appropriate results relative to the actual data.

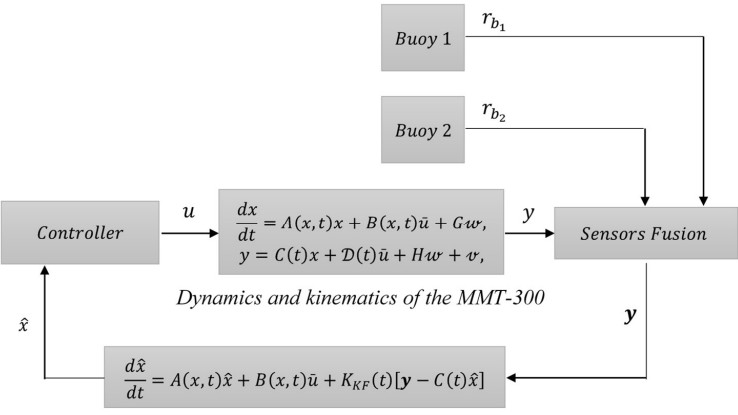

**Figure 10.** Positioning system block diagram.

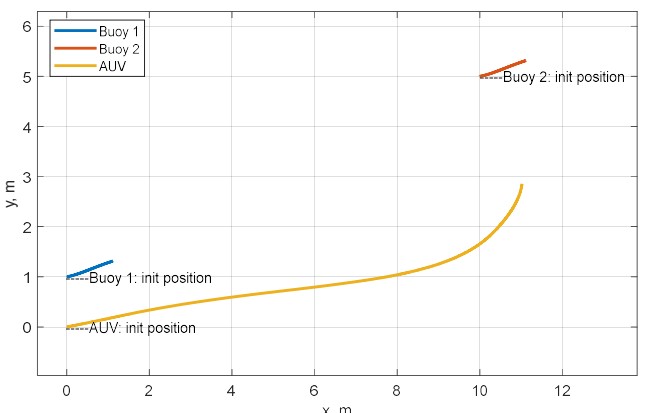

**Figure 11.** XY positions of buoys and AUV.

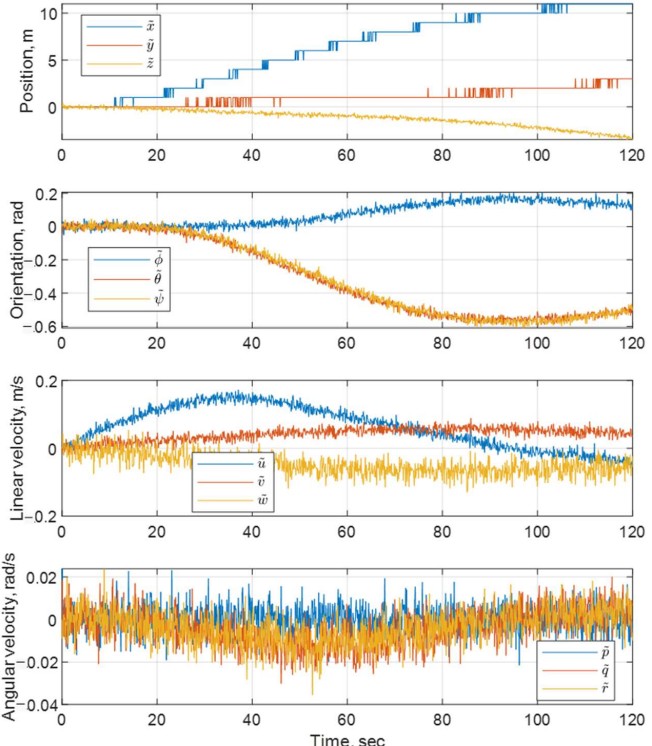

**Figure 12.** Measurement charts.

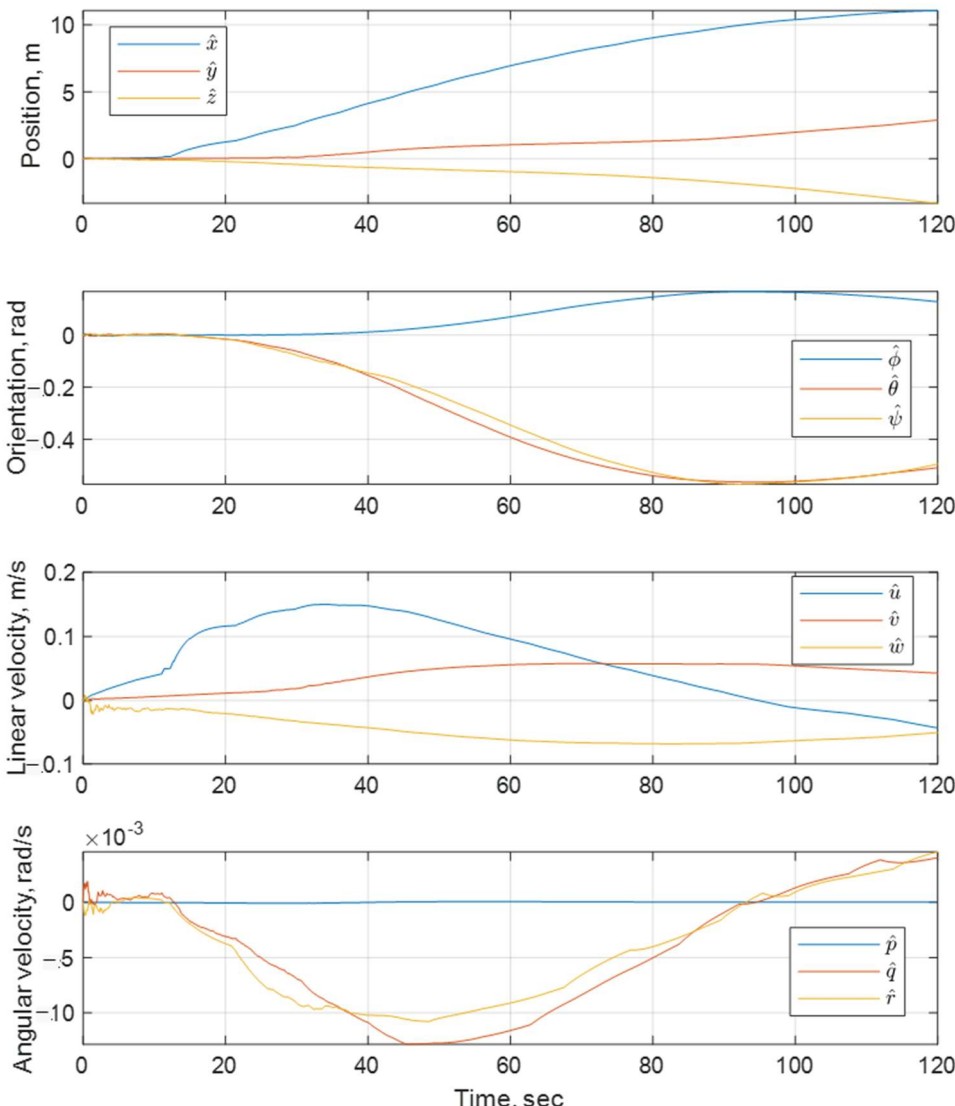

**Figure 13.** EKF estimation charts.

A more correct assessment of the filter performance can be made by the error with respect to the real data (Figure 14). Quantitative evaluation is already defined by quality metrics. MSE, MAE, and RMSE metrics are summarized in Table 7.

**Table 7.** Quality metrics.

| Metric | Value |
|---|---|
| MSE ($\eta$) | $\begin{bmatrix} 1.868327 & 3.291973 & 0.390871 & 0.030260 & 0.044263 & 0.318385 \end{bmatrix}^T$ |
| MSE ($\nu$) | $\begin{bmatrix} 0.191705 & 0.115427 & 0.063261 & 0.002418 & 0.043486 & 0.013741 \end{bmatrix}^T$ |
| MAE ($\eta$) | $\begin{bmatrix} 0.029913 & 0.082423 & 0.008349 & 0.001244 & 0.002044 & 0.004935 \end{bmatrix}^T$ |
| MAE ($\nu$) | $\begin{bmatrix} 0.003227 & 0.001675 & 0.004679 & 0.000030 & 0.000247 & 0.000525 \end{bmatrix}^T$ |
| RMSE ($\eta$) | $\begin{bmatrix} 0.074581 & 0.116712 & 0.139543 & 0.000732 & 0.003981 & 0.016776 \end{bmatrix}^T$ |
| RMSE ($\nu$) | $\begin{bmatrix} 0.007249 & 0.004172 & 0.002344 & 0.000038 & 0.000389 & 0.000886 \end{bmatrix}^T$ |

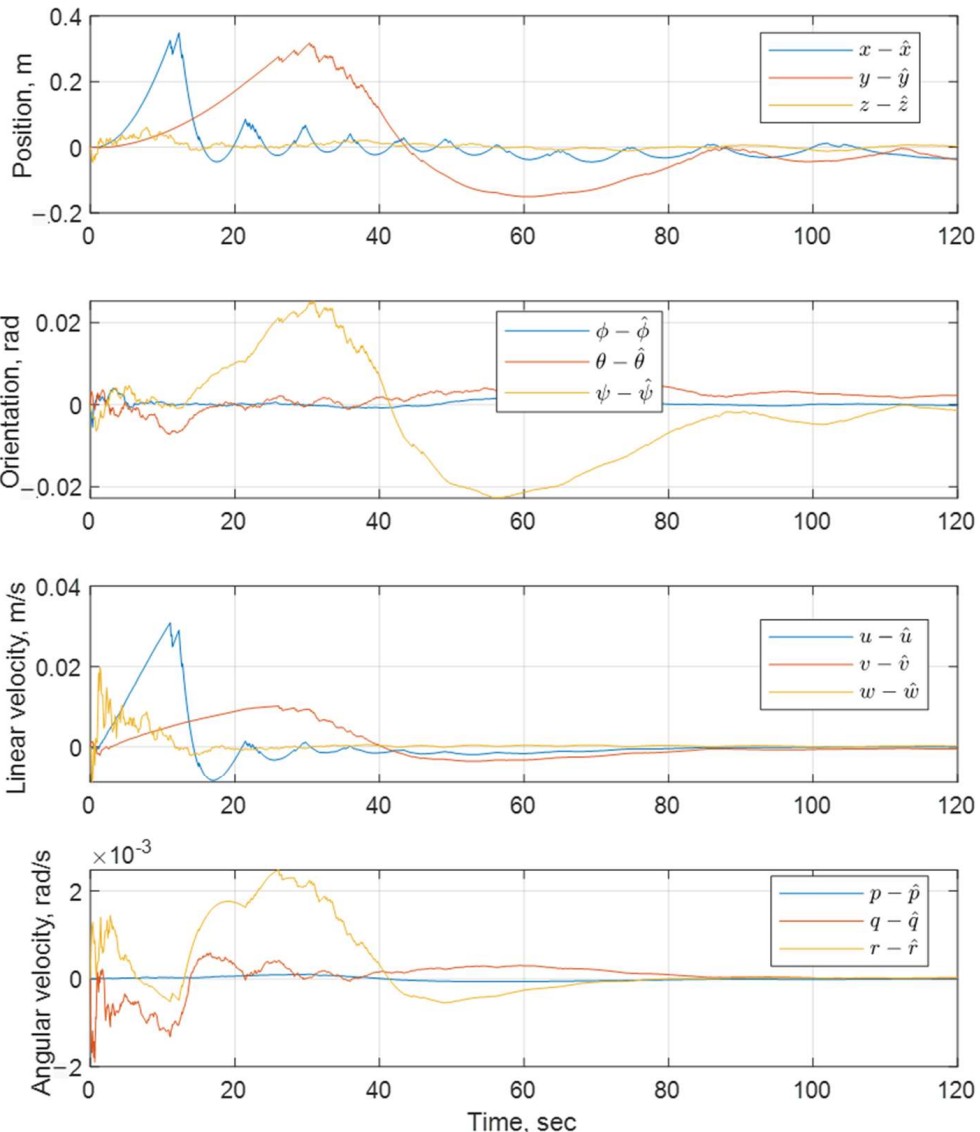

**Figure 14.** EKF estimation error charts.

## 5. Discussion

The main goal of this paper was to develop a high-quality autonomous navigation performance using MEMS-sensor-based IMUs with a rough accuracy class, a Doppler log, and a pressure sensor. During this study, a simulation model of underwater robot motion was developed. All proposed algorithms and approaches were tested with simulation modeling tools and using natural data.

The presented position estimation graphs showed that the filtering of noisy data is fully realized. On the considered time interval, the error between the measured value and the actual value tends toward zero. The above judgment is supported by the numerical results of the given quality metrics. The finest results were obtained in restoring the velocities of the underwater vehicle.

The specified goal was achieved: it was shown that it is practically possible to achieve an acceptable quality of navigation in autonomous mode using a class of sensors with rough accuracy. The positioning error turns out to be limited over a long time interval, even if the motion starts from a large area of initial uncertainty. The outcomes were supported by conducted field experiments, which were carried out in the Black Sea water area with the navigation coordinate system origin at longitude 33.46° and latitude 44.58°.

## 6. Conclusions

This paper has presented a positioning method to utilize the information integration mechanism for various onboard sensors using EKF and MRC techniques. The error plots and numerical quality metrics showed that the system is able to recover the AUV position values with high accuracy.

At the same time, the method is not lacking in disadvantages. The duality of the position determination problem is preserved. This fact allows us to avoid USBL systems that provide information about bearing angles but sets certain limitations since knowledge of the previous position is required. Moreover, the EKF proposed for combining the readings is rather demanding on computational resources.

Further research will be focused on the generation of control laws for a swarm of submersibles relative to the leader since the main problem of cooperative navigation and positioning still remains state estimation and accurate collection of observations, especially for the leading element.

**Author Contributions:** Conceptualization, A.K.; methodology, A.K. and V.K.; software, K.D.; validation, A.K., K.D., and V.K.; formal analysis, A.K.; investigation, A.K. and K.D.; resources, A.K.; data curation, V.K.; writing—original draft preparation, A.K. and K.D.; writing—review and editing, A.K.; visualization, K.D.; supervision, V.K.; project administration, A.K.; funding acquisition, A.K. All authors have read and agreed to the published version of the manuscript.

**Funding:** This research received no external funding.

**Institutional Review Board Statement:** Not applicable.

**Informed Consent Statement:** Not applicable.

**Data Availability Statement:** The data presented in this study are available upon request from the corresponding author.

**Conflicts of Interest:** The authors declare no conflict of interest.

## Abbreviations

The following abbreviations are used in this manuscript:

| | |
|---|---|
| GPS | Global Positioning System |
| DVL | Doppler Velocity Logger |
| IMU | Inertial Measurement Unit |
| PS | Pressure Sensor |
| AUV | Autonomous Underwater Vehicle |
| HCS | Hydroacoustic Communication System |
| HNCS | Hydroacoustic Navigation and Communication System |
| HANS | Hydroacoustic Navigation System |
| LBL | Long Baseline |
| SBL | Short Baseline |
| USBL | Ultra-Short Baseline |
| KF | Kalman Filter |
| EKF | Extended Kalman Filter |
| OCNS | Onboard Control and Navigation System |
| PSC | Propulsion–Steering Complex |
| MEMS | Microelectromechanical System |
| MRC | Maximum-Ratio Combining |
| RMS | Root Mean Square |
| MSE | Mean Squared Error |
| MAE | Mean Absolute Error |
| RMSE | Root-Mean-Square Error |

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
