# Peer review of "Autonomous Underwater Vehicle Navigation via Sensors Maximum-Ratio Combining in Absence of Bearing Angle Data"

_jmse, doi:10.3390/jmse11101847_

Round 1

Reviewer 1 Report

The abstract should be re-written to highlight the quantitative findings of the research.

Many pages introduce the model of underwater vehicles. The corresponding context is well-known and should be simplified.

f_e in (3) denotes the vector of external disturbances. Does it act on the body frame? Or in the inertial frame?

What are emergency sensors and actuators?

Please add the unit of buoyancy center vector.

For through-body thrusters in different parts of the underwater body, the characteristics are different from each other. Eq. (13) should be given more remarks.

B_UAPS in (32) is not defined.

The outcomes were supported by field experiments. Please give more details of field experiments including original data.

Please double check the text, i.e., the possible positions of the ‘submersible’ are

A paragraph should contain more than one sentence.

Author Response

Point 1: The abstract should be re-written to highlight the quantitative findings of the research.

 Response 1: Thank you for your precious comments! Justification of the present paper's value to researchers was added to the abstract.

Point 2: Many pages introduce the model of underwater vehicles. The corresponding context is well-known and should be simplified.

 Response 2: Thank you for your precious comments! A more detailed description of underwater vehicle kinematics has been removed from the paper, with the basic equation and references retained, and the expanded forms of the dynamics matrices have been removed.

 Point 3: f_e in (3) denotes the vector of external disturbances. Does it act on the body frame? Or in the inertial frame?

 Response 3: Thank you for your precious comments! In fact, this vector is expressed in the inertial frame. Appropriate remarks were added in the equation's description.

 Point 4: What are emergency sensors and actuators?

 Response 4: Thank you for your precious comments! Details have been added to the article.

 Point 5: Please add the unit of buoyancy center vector.

 Response 5: Thank you for your precious comments! Units for the center of mass and center of buoyancy vectors were added.

 Point 6: For through-body thrusters in different parts of the underwater body, the characteristics are different from each other. Eq. (13) should be given more remarks.

 Response 6: Thank you for your precious comments! The static characteristic of all thrusters is the identical for the presented underwater vehicle. The equation describing the forces and moments generated by the thrusters already gives these characteristics through trigonometric expressions determined by the geometric layout of the thrusters.

 Point 7: B_UAPS in (32) is not defined.

 Response 7: Thank you for your precious comments! The variable naming in the equation has been corrected

 Point 8: The outcomes were supported by field experiments. Please give more details of field experiments including original data.

 Response 8: Thank you for your precious comments! A more detailed description of the used hardware was given and details of the experiments were disclosed.

Reviewer 2 Report

This work presents a maximum ratio combining sensor fusion scheme using an extended Kalman filter for underwater vehicle positioning. Communication devices (buoys) giving location using a slant range mechanism, inertial sensors, a Doppler log, and a pressure sensor are used for the implementation. The results of simulation modeling are presented, and the outcomes were supported by field experiments.

Some issues should be addressed before this manuscript can be considered for publication.

1) Full numerical simulation and field experiments details must be provided so that the results can be reproduced. In sections 3 and 4, detailed implementation information should be provided (hardware, software, configuration, settings).

2) The field experiments implementation and results should be properly presented.

3) Section numbering should be corrected (3.4.2., 3.4.3., 3.4.4., 2.2., “4. Discussion”).

4) The discussion section, at the end of section 4, should provide a valuable analysis of the results.

5) What are the limitations and issues to address for the practical implementation of this method?

6) A Conclusion section should be provided to include quantitative results, advantages and disadvantages, limitations, and recommendation for the implementations. Recommendations and guidelines for future work should be provided.

Author Response

Response to Reviewer 2 Comments

Point 1: Full numerical simulation and field experiments details must be provided so that the results can be reproduced. In sections 3 and 4, detailed implementation information should be provided (hardware, software, configuration, settings).

 Response 1: Thank you for your precious comments! Hardware details including USBL system and operator computer specifications were added to the work.

Point 2: The field experiments implementation and results should be properly presented.

 Response 2: Thank you for your precious comments! Details of the used equipment and the experiment location have been added to the paper.

 Point 3: Section numbering should be corrected (3.4.2., 3.4.3., 3.4.4., 2.2., “4. Discussion”).

 Response 3: Thank you for your precious comments! The numeration of all sections and subsections has been corrected.

 Point 4: The discussion section, at the end of section 4, should provide a valuable analysis of the results.

 Response 4: Thank you for your precious comments! Analysis of cited metrics and error plots has been added to the work.

 Point 5: What are the limitations and issues to address for the practical implementation of this method?

 Response 5: Thank you for your precious comments! Limitations of the method have been described in the paper and highlighted in the Conclusion section.

 Point 6: A Conclusion section should be provided to include quantitative results, advantages and disadvantages, limitations, and recommendation for the implementations. Recommendations and guidelines for future work should be provided.

 Response 6: Thank you for your precious comments! A Conclusion section has been included in the paper to describe the obvious advantages and disadvantages of the proposed positioning method.

Reviewer 3 Report

Comments and suggestions can be found in the attachment.

Usually the articles will use some specialized words in the field of study, such as this article is in the field of underwater robot navigation, please note which words are used. Sometimes more than one word can express the same meaning, but we should choose the word whose meaning is most commonly used in the field of navigation.

Author Response

Response to Reviewer 3 Comments

Point 1: In paragraph 7 of section 1, the information obtained by PS is usually the depth of the water, rather than the altitude written in the present article.

 Response 1: Thank you for your precious comments! The terminology has been corrected.

Point 2: LBS is mentioned in paragraph 8 of section 1, and LBL appears in paragraph 9 of section 1. There are problems with the expression of nouns.First of all, the author wants to describe the term should be underwater acoustic positioning system; Secondly, according to the length of the acoustic baseline, the underwater acoustic positioning system can be divided into LBL, SBL and USBL. Please pay attention to the terminology.

 Response 2: Thank you for your precious comments! The abbreviations for the various baselines have been unified.

 Point 3: In paragraph 14 of section 1,"especially for the lead vehicle in the group", the object of this study is a single AUV, let alone the content of cooperative navigation of multiple AUVs.

Response 3: Thank you for your precious comments! Cooperative navigation is now mentioned exclusively in the Discussion section when defining a vector for further research.

 Point 4: In section 2.1,what are the physical quantities represented by v1, v2 ,u, v,w, p, q,r in formula (2)? Please specify or explain.

 Response 4: Thank you for your precious comments! The notation for these variables is common to a number of sources. A link has been included in the article where each variable is described in more detail.

 Point 5: In formula (5), there is a . The translation matrix from the navigation coordinate system to the carrier coordinate system should be stated in the explanation, and the navigationcoordinate system is usually represented by the Angle symbol N.

 Response 5: Thank you for your precious comments! An appropriate remark was left when describing the variable. The notation has been retained to match the referenced source.

 Point 6: In section 2.2.1,the last item should be Doppler velocity logger.

 Response 6: Thank you for your precious comments! The definition has been corrected.

 Point 7: In section 2.2.3,what "GANS" stands for,?And where various abbreviations first appear in the text, their complete names and their meanings should be indicated.

 Response 7: Thank you for your precious comments! The abbreviation GANS has been corrected to HANS. Furthermore, a section for defining abbreviations has been added.

 Point 8: In Table 3,what does "mg,N" mean relative to "buoyancy,B"? Please give a reasonable explanation or correction. Please give a reasonable explanation or correct it.

 Response 8: Thank you for your precious comments! Buoyancy forces are measured in Newtons, for this reason the given notation was used.

 Point 9: The marking format of the physical quantity in the formula in the whole paper is not uniform, for example, the marking format of formula (3) and formula (14) is not agreed. Please be consistent throughout the text.

 Response 9: Thank you for your precious comments! The used equations are used in the cited sources in the presented matrix form. For clarification, the dimensions of the parameter were explicitly specified in equation (3).

 Point 10: In section 2.3.4 and 2.3.5,whether the content can be merged? Please give a reasonable explanation or correct it.

 Response 10: Thank you for your precious comments! Section 2.3.5. has been renamed to reflect its content.

 Point 11: In the physical quantity interpretation of formula (28),"  "Should be changed to"  ". Please give a reasonable explanation or correct it.

 Response 11: Thank you for your precious comments! The notation in describing the equation has been corrected.

 Point 12: The title "Inertial sensors" in section 2.5.1 is the same as it is in section 3.4.2 below, which is related to depth and should be changed. At the same time, there is a problemwith the title serial number. After "2.5.1", there are "3.4.2,、3.4.3、 3.4.4", please sort out in the full text.

 Response 12: Thank you for your precious comments! The semantic content of section names has been corrected. Sections numbering has also been corrected.

 Point 13: In formula (31), in the modeling of DVL, why there are two physical quantities " " and" "at the same time. Please give a reasonable explanation.

 Response 13: Thank you for your precious comments! The Doppler velocity logger measures velocity in two axes only, for this reason the notation is presented in a such form.

 Point 14: In the physical quantity interpretation of formula (32)," "Should be changed to"  ".Please give a reasonable explanation or correct it.

 Response 14: Thank you for your precious comments! The variable naming has been corrected

Point 15: In section 3,entitled "Navigation system components", the various navigation methods are introduced conceptually, and the content here should be specific to the experimental object AUVMMT-300, so there is no need to spend space on these navigation methods.These navigation methods should appear in section 1, "1 Introduction".The third part of the paper should fully describe the specific methods adopted by the research, highlighting the words "inAbsence of Bearing Angle Data" and "maximum ratio combination" in the title of the article, and express their specific content in detail. Please give a reasonable explanation or or correct it.

 Response 15: Thank you for your precious comments! An overview of navigational methods has been moved to the Introduction section. The corresponding section describing the proposed system already describes the specifics.

 Point 16: In Figure 9,there are two "Possible positions" . How to determine which one the AUV is in? Please give a reasonable explanation.

 Response 16: Thank you for your precious comments! The solution to this issue is presented in the sentence "The duality of solving the equation is resolved if the previous underwater vehicle position is known"

 Point 17: In the abstract of the article, it says "giving location using a slant range mechanismin absence of bearing angle data, inertial sensors, a Doppler log, and a pressure sensor", but the data is still used in the simulation later in the article.For example, in the title "Slant ranges usage", depth is still used.Please give a reasonable explanation.

 Response 17: Thank you for your precious comments! The abstract word order has been corrected. Only the bearing angle is not actually used.

Reviewer 4 Report

A very good research with particularity suggestive results presented in the paper. More tests and the identification of specific applications are recommended.

Author Response

Response to Reviewer 4 Comments

Point 1: A very good research with particularity suggestive results presented in the paper. More tests and the identification of specific applications are recommended.

 Response 1: Thank you for your precious comments! A Conclusion section has been added to the paper which additionally reflects the potential applications of the proposed techniques.

Round 2

Reviewer 1 Report

My first-round comments have been well addressed.

None

Reviewer 2 Report

The authors addressed the recommendations, the manuscript has been sufficiently improved and could be considered for publication.